# Reversible Data Hiding in Encrypted Image Using Multiple Data-Hiders Sharing Algorithm

**DOI:** 10.3390/e25020209

**Published:** 2023-01-21

**Authors:** Chi-Yao Weng, Cheng-Hsing Yang

**Affiliations:** Department of Computer Science and Artificial Intelligence, National Pingtung University, Pingtung 900, Taiwan

**Keywords:** encrypted image, reversible data hiding, secret sharing, multi-hider mechanism

## Abstract

Reversible Data Hiding in Encrypted Image (RDHEI) is a technology for embedding secret information in an encrypted image. It allows the extraction of secret information and lossless decryption and the reconstruction of the original image. This paper proposes an RDHEI technique based on Shamir’s Secret Sharing technique and multi-project construction technique. Our approach is to let the image owner hide the pixel values in the coefficients of the polynomial by grouping the pixels and constructing a polynomial. Then, we substitute the secret key into the polynomial through Shamir’s Secret Sharing technology. It enables the Galois Field calculation to generate the shared pixels. Finally, we divide the shared pixels into 8 bits and allocate them to the pixels of the shared image. Thus, the embedded space is vacated, and the generated shared image is hidden in the secret message. The experimental results demonstrate that our approach has a multi-hider mechanism and each shared image has a fixed embedding rate, which does not decrease as more images are shared. Additionally, the embedding rate is improved compared with the previous approach.

## 1. Introduction

Multimedia security technology is used to prevent unauthorized users from copying, sharing, and modifying media content. To prevent this problem, encryption and information hiding are often used to protect media content. As far as information hiding technology is concerned, traditional information hiding technology will destroy the content of the cover image. However, in some exceptional cases, such as military, medical, and legal document images, the slight distortion of the image is entirely unacceptable. Therefore, whether these images can be completely restored is very important. Reversible data hiding scheme (RDH) can correspond with the requirement of being lossless. RDH methods applied the methodology of changing context to hide the secret data in cover media. After data extracting, the changing context will be fully recovered to the cover media. On the other hand, RDHEI (Reversible Data Hiding in Encrypted Images) technology can combine encryption technology with RDH technology, which can not only hide secret information in the image, but can also encrypt the image to protect the image content.

RDH techniques can be broadly classified into three types: (1) Difference Expansion [1,2,3]: Difference expansion is performed by expanding the difference between adjacent pixels, and then secret information is embedded into the difference. Since the difference will be expanded after the secret information is embedded, this technique will inevitably produce a larger distortion; (2) Histogram Shift [4,5,6]: through Histogram Shift, the histogram is shifted by the original image or the histogram of the predicted discrepancy, and the empty position after the shift is used to embed the secret information; (3) Lossless Compression [7,8]: The secret information is hidden in the extra space after compressing the original image. Since lossless compression may lead to significant degradation of visual quality, it has received less attention. By and large, the RDH approach is based on the above with some other added strategies. For example, in 2019, Zhao et al. proposed a method of histogram displacement [9]. First, the secret information is converted into a message with only −1, 0, and 1 by encoding. Then, the middle segment bin is selected for embedding on the histogram displacement. These bins can all be used to embed a −1, 0, or 1 message by shifting them left and right. Finally, the band size is adjusted using the Threshold, and the histogram is created using the prediction error values generated by the Chess Board Prediction method. In 2021, Gao et al. proposed a histogram displacement method for embedding medical images [10]. By dividing medical images into ROI and NROI, the embedding process will stretch the histogram created by the pixel values of ROI to the left and right, which expands the embedding capacity of ROI and enhance the contrast of the image.

RDHEI techniques can be divided into two categories: (1) Vacating Room before Encryption, VRBE [11,12,13,14,15,16,17]: the original image is first vacated and then encrypted; and (2) Vacating Room after Encryption, VRAE [18,19,20,21]: The image is first encrypted, and because the encrypted image retains certain properties, the encrypted image can be hidden. VRBE technology mainly focuses on reversible information hiding or compression of the original image to free up space. It encrypts the image, so there is space to hide information in the encrypted image. For example, in 2013, Ma et al.’s RDHEI method belongs to the technology of Vacating Room before Encryption [11]. They performed reversible information hiding in the original image beforehand and embedded the LSB part of a block into the image. As a result, the encrypted image can use the LSB part of this block to hide data. In 2018, Puteaux et al. performed MSB prediction and compression on the original image so that the MSB part could vacate most of the space, and the encrypted image could use this MSB part to embed the data [13]. In 2021, Yin et al. proposed a method based on pixel prediction and multi-MSB plane, which belongs to Vacating Room before Encryption [17]. First, it used the MED predictor to obtain the predicted value, and to calculate the predicted error (PE) between the predicted value and original value. Next, the sign bit of PE is stored in bit-plane 8, and the absolute value of PE is represented in bit-plane 7 (bit-plane 7) and bit-plane 1 (bit-plane 1). The bit plane is then divided into Uniform Blocks and Non-Uniform Blocks, and these blocks are rearranged. Since prediction errors are usually concentrated around zero, these bit planes, which have a large number of uniform blocks, can be encoded and compressed to free up space. On the other hand, VRAE technology mainly uses a specific encryption technique to maintain the dependency of the neighboring pixels in the encrypted image. Therefore, this feature can be used to hide the information in the encrypted image. Commonly used encryption techniques include Block-Level Scrambling [18] or Block-Level Encryption [19], in which pixels in the same block use the same random value for XOR (Exclusively-OR). For example, in 2019, Qin et al. proposed a VRAE method [18] that first encrypts the image by stirring the image between bit-planes, between blocks, and within blocks so that the pixels within the blocks retain their dependency. Then, block classification and encoding compression are used to generate the embedding space when embedding information. In 2019, Wang et al. proposed a VRAE approach, where their encryption uses inter- and intra-block stirring so that pixels within a block retain dependency [19]. Then, for the information embedding, they use the 3-LSB of all pixels in the flip block to divide the block into EB (Embeddable Block), and NEB (Non-Embeddable Block), and only the secret information is embedded in the EB. The location map of the block is embedded using the RDH method to hide 5-MSB of all pixels.

Recently, some scholars have proposed the RDHEI approach based on Secret Sharing [20,21,22]. In 2018, Wu et al. proposed an RDHEI method based on Secret Sharing [20]. However, their encrypted image size was more than twice the size of the original image. In 2019, Chen and other scholars proposed a secret sharing-based RDHEI method that converts original images into encrypted images via the sharing technique of Shamir’s Secret Sharing and distributes the encrypted images to information hiders for information hiding [21]. Since the size of the generated encrypted image is the same as that of the original image, no information inflation occurs. All of the above methods are targeted at a single information supporter. In 2020, Chen and other scholars proposed a new secret-sharing RDHEI approach [22], which expands the original single information hider into multiple information hiders. The total embedding volume of Chen et al.’s method is fixed, so the embedding rate decreases as the number of shared images increases.

In this study, we propose a new RDHEI technique where we hide the pixel values into the polynomial coefficients by dividing multiple pixels into a group. First, the image owner confidentially shares the image, generating a shared image and leaving space in each shared image for information embedding. Then, the information recipient can embed a confidential message after receiving the shared image. Finally, the recipient can decrypt the image and remove the hidden information. The recipient needs to obtain at least a Threshold before the original image can be recovered. Our approach enables multiple information hiders and has a more suitable fixed embedding rate. Furthermore, it solves the problem that the embedding rate becomes smaller as the number of shared images becomes larger and successfully improves the embedding rate compared with the previous method.

The remaining sections of this study are organized as follows. Section 2 reviews Chen’s existing work [22]. Section 3 describes our novel secret sharing images with multiple data hiders method. This is followed by its experimental results along with the analysis and the existing works comparisons in Section 4. Finally, the conclusion is drawn in Section 5.

## 2. Preliminary

This section introduces Chen et al.’s approach, which develops RDHEI techniques through Shamir’s Secret Sharing method. Chen et al.’s method has the role of multiple information hiders, but their total embedding is fixed, so the embedding rate decreases as the number of shared images increases. Chen et al.’s approach is introduced in the following.

Figure 1 shows the flowchart of encryption, embedding, extraction, and decryption by Chen et al. First, the content-owner encrypts the image and uses the key *ke* to encrypt the original image into *n*. Each encrypted image is the same size as the original image. Then, each encrypted image is assigned to a different information hider. Each information hider, Data-hider *t*, hides the confidential information in the encrypted image *t* through the key *kh_t_*. Finally, the 1≤t≤n receiver simply collects any *k* encrypted images with embedded information and related keys to extract the embedded information and decrypt it to obtain the original image.

The content-owner performs the image encryption by first dividing image *I* into parts A and B. Part A is the first 10 pixels of the image, and the remaining part is part B. Figure 2 is the representation. Part A is used to store the parameters of the information hiding and the parameters of the histogram displacement process. First, the following bits are extracted from part A: all bits of the first two pixels and 3 LSBs of the last 8 pixels. Next, all the extracted bits are defined as AP. Then, scan all pixels in the part B, recording the location of the pixel values ≥ 250 with the location map, modify these pixel values to 249, and create a histogram of part B. From the histogram, we found the most suitable embedding point for embedding the location map and AP, which was defined as PP. In the histogram, all the values between o PP + 1 and 249 are shifted one place to the right so that the location map and AP can be using PP for 0 and for 1, as shown in Figure 3. Finally, the PP value is embedded into the 3-LSB part of the last 8 pixels of the A part. After the above pre-processing, image *I* is modified to image *I′*.

Using Equation (1), the B part of the image *I′* is shared confidentially via Shamir’s Secret Sharing, where *T_i_*_,*j*_^(0)^ is a constant, *a_i_*_,*j*_^(*α*)^ is an integer random number, and *I′_i,j_* is the pixel value of position (*i*,*j*). The image owner uses key *ke* to generate *n* non-zero random integers *x_i_*_,*j*_^(*t*)^, t=1, 2, …,n, to bring *x_i_*_,*j*_^(*t*)^ into Equation (1) within *x* to obtain the confidential sharing result *F_i_*_,*j*_ (*x_i_*_,*j*_^(*t*)^), t=1, 2, …,n. It then hides the parameters *t* and *n* into the first two pixels of part A. From this we can obtain *n* encrypted images *E*^(*t*)^, t=1, 2, …,n.
(1)$$F_{i,j}\left(x\right)=\left\{\begin{array}{c} \left(T_{i,j}^{\left(0\right)}+{I^{\prime }}_{i,j}x\right)\;\mathrm{mod}\;p,\;\;\;\;\;\;\;\;\;\;\;\;\;\\ \left(T_{i,j}^{\left(0\right)}+{I^{\prime }}_{i,j}x+\overset{k-1}{\underset{a=2}{\sum }}a_{i,j}^{\left(a\right)}x^{a}\right)\;\mathrm{mod}\;p, \end{array}\begin{array}{c} \mathrm{if}\;k=2,\\ \;\;\;\;\mathrm{if}\;2<k\leq n \end{array}\right.$$

Data embedding is performed using the *n* information hiders separately. The *t*-th information hider obtains the shared encrypted image *E*^(*t*)^ by scanning the first 2 pixels to obtain the *t* and *n* values. It divides the pixels into groups. Each group contains *n* pixels and replaces the *l*
$\left(1\leq l\leq 7\right)$ LSB of the *t*-th pixel in each group with the secret information. As a result, the *n* encrypted images *Em*^(*t*)^, t=1, 2, …,n with the secret information embedded can be obtained. The embedding rate of this method is $\frac{\lfloor \frac{WH-2}{n}\rfloor \;\cdot l}{WH}\approx \frac{l}{n}$ bpp (bit per pixel), where *W* and *H* are the width and height of the image, respectively.

In this part of information extraction and image decryption, it is assumed that the first *k* shared images are *Em*^(*t*)^, t=1, 2, …,k and the key is *kd*. In the information extraction phase, for any shared image *Em*^(*t*)^, t=1, 2, …,k. By sharing the first 2 pixels of the image, *t* and *n* can be obtained. Next, divide all the pixels in the B part of *Em*^(*t*)^ into a group, and extract the *l*-LSB of the *t*-th pixel in each group to obtain the secret information of the shared image *Em*^(*t*)^. In image decryption, for the pixel *Em*^(*t*)^*_i_*_,*j*_, t=1, 2, …,k at the coordinates (*i*,*j*), and the non-zero random integers *x_i_*_,*j*_^(*t*)^, and t=1, 2, …,k generated by the key *kd*, equation can be obtained through the Lagrange polynomial. (1) Polynomial *F_i_*_,*j*_(*x*) = $\sum_{t=1}^{k}\;\left(Em^{\left(t\right)}{}_{i,j}\right.\;\overset{k}{\underset{a=1,a\neq t}{\prod }}\frac{x-X_{a}}{X_{t}-X_{a}})\;\mathrm{mod}\;251$, where *X_a_* is substituted using *x_i_*_,*j*_^(*a*)^. However, *Em*^(*t*)^*_i_*_,*j*_, t=1, 2, …,k may not be able to obtain the correct *F_i_*_,*j*_(*x*) because *l* bits are hidden into *l*-LSB. Without a loss of generality, assuming that the first pixel to share an image is hidden in *l*-bit, i.e., *Em*^(1)^*_i,j_* is hidden in *l*-bits, then when the *Em*^(1)^*_i_*_,*j*_ of *l*-LSB is modified to 0, 1, 2, ..., 2*^l^* − 1, one of them must be *Em*^(1)^*_i_*_,*j*_ before embedding *l*-LSB, which is called the correct *Em*^(1)^*_i_*_,*j*_. The correct *Em*^(1)^*_i_*_,*j*_ can be identified by using *F_i_*_,*j*_ (0) = *T_i_*_,*j*_
^(0)^, where *T_i_*_,*j*_
^(0)^ is the constant used in the encryption step. After constructing *F_i_*_,*j*_ (*x*), from Equation (1) pixel *I′_i_*_,*j*_ of the coordinates (*i*,*j*) can be obtained and therefore the image *I′* can also be obtained. Finally, the image *I′* is restored to the original image *I* using the information in part A.

## 3. Proposed Method

In this section, we propose a new approach to the interplay of confidential image sharing and information hiding. Our approach is also based on Shamir’s Secret Sharing technique of confidential sharing. The traditional Shamir (*k*, *n*)-threshold secret sharing technique shares confidential information *D* into *n* shared information and the original confidential information *D* can be calculated by taking out *k* of the shared information. This method requires the construction of a *k*−1 polynomial, as follows:(2)$$q\left(x\right)=a_{0}+a_{1}x+\ldots +a_{k-1}x^{k-1}\;\mathrm{mod}\;p$$
where $a_{0}=D$, p is a prime, constant, or coefficient $a_{i}<p,\;i=0,\;1,\;\ldots ,\;k-1$. If this mechanism is applied to image sharing, for example, for a grayscale image with a gray scale of 256, to share a pixel value *P*, the above polynomial is usually used with $a_{0}=P$ and p = 251. Because the value range of pixel value *P* is 0~255, all pixel values of the image must be pre-processed so that the range of all pixel values becomes 0~250.

We propose the finite field ($GF\left(2^{n}\right)$) strategy, modifying Shamir’s polynomial formulation, as follows:(3)$$q\left(x\right)=a_{0}+a_{1}x+\ldots +a_{k-1}x^{k-1}\;$$
where the constants or coefficients are $a_{i}\in GF\left(2^{8}\right),\;i=0,\;1,\;\ldots ,\;k-1$ and both multiplication and addition are performed using the $GF\left(2^{n}\right)$ operator. Since the $GF\left(2^{n}\right)$ quadrature is closed, i.e., if $a,b\in GF\left(2^{n}\right)$, then $a+b\in GF\left(2^{n}\right)$, $a-b\in GF\left(2^{n}\right)$, $a\times b\in GF\left(2^{n}\right)$, and $a/b\in GF\left(2^{n}\right)$, the polynomial can be directly applied to image sharing, and in the case of grayscale images with 256 (=$2^{8}$) gray levels, no pixel-specific pre-processing is required at all.

Our method has three roles: Content-owner, Data-hiders, and Receivers, as shown in Figure 4, which shows the operation flow of our method in image encryption, data embedding, data extracting, and image decryption. We take Shamir’s (*k*, *n*)-threshold as an example, where *n* means *n* shared images are generated and *k* means *k* shared images are obtained to decrypt the original image. First of all, the image owner can generate *n* shared images through Shamir’s Secret Sharing technology and Galois field 256 for confidential sharing of the original image *M*. The key $X_{1},\;\ldots ,X_{n}$ are the keys required to generate $C^{\left(1\right)},\;C^{\left(2\right)},\;\ldots ,C^{\left(n\right)}$, respectively. Next, the data-hider has a separate pair of *n* shared images to embed the confidential data *S* ^(*n*)^ embedded in *n* in a shared image to generate an embedded shared image ${C^{\prime }}^{\left(1\right)},\;{C^{\prime }}^{\left(2\right)},\;\ldots ,{C^{\prime }}^{\left(n\right)}$. Lastly, *n* image recipients receive the embedded shared image, respectively ${C^{\prime }}^{\left(1\right)},\;{C^{\prime }}^{\left(2\right)},\;\ldots ,{C^{\prime }}^{\left(n\right)}$. Any image recipient with an embedded key $X_{1},\;\ldots ,X_{n}$ can retrieve the confidential information embedded by the data-hider from the shared image. In addition, as long as any *k* of the embedded shared images can be decrypted, assuming that the *k* embedded shared images are ${C^{\prime }}^{\left(1\right)},\;\ldots ,{C^{\prime }}^{\left(k\right)}$ then using the key $X_{1},\;\ldots ,\;X_{k}$ can decrypt the original image *M* and retrieve confidential data *S*. Linear operations on polynomials through a Galois domain, whose addition and multiplication are similar to normal addition and multiplication, except that the result of the operations are elements of the domain.

### 3.1. Image Encryption

For the image encryption part, we use Shamir’s confidential sharing and pixel grouping strategy. Figure 5 shows our confidential sharing process. We group the original pixels, if for Shamir’s sharing (*k, n*)-threshold, each group contains *k* pixels, in the sharing process of group (*P_i_*_,*j*_, …, *P_i_*_,*j*+*k−*1_), we use these pixel values to build *k* − 1 times polynomial, then use *X*_1_, *X*_2_, …, *X_n_* substituted into the polynomial to obtain the shared pixels *y*^(1)^*_i_*_,*j*_, …, *y*^(*n*)^*_i_*_,*j*_. Each shared pixel, *y*^(*t*)^*_i_*_,*j*_, can be split into *k* pixels to produce *C*^(*t*)^*_i_*_,*j*_, …, *C*^(*t*)^*_i_*_,*j*+*k−*1_. Since the group (*C*^(*t*)^*_i_*_,*j*_, …, *C*^(*t*)^*_i_*_,*j*+*k−*1_) holds only *y*^(*t*)^*_i_*_,*j*_ with 8 bits, so the group can be empty of $8k-8=8\left(k-1\right)$ bits of space.

Here is our cryptographic algorithm. Our Algorithm 1 is illustrated with Shamir (*k*, *n*)-thresholds, so the polynomial is a (*k −* 1)-*th* polynomial.
**Algorithm 1:** Image Sharing
**Input:** Image *M*, Encryption keys *X_t_*, t=1, 2, …, n.**Output:** Sharing images *C*^(*t*)^, t=1, 2, …, n.Step 1: A set of pixels (*P_i_*_,*j*_, …, *P_i_*_,*j*+*k−*1_) is extracted from Image *M.* The polynomial $P_{i,j+k-1}x^{k-1}+P_{i,j+k-2}x^{k-2}+\ldots +P_{i,j}$ is generated using the set of pixels (*P_i_*_,*j*_, …, *P_i_*_,*j*+*k−*1_).Step 2: Substitute the Encryption key *X_t_*, t=1, 2, …, n through 256 Galois field into the polynomial $P_{i,j+k-1}x^{k-1}+P_{i,j+k-2}x^{k-2}+\ldots +P_{i,j}$ to obtain *n* encrypted pixel values *y*^(*t*)^*_i_*_,*j*_, where *y*^(*t*)^*_i_*_,*j*_
$=P_{i,j+k-1}x_{t}{}^{k-1}+P_{i,j+k-2}x_{t}{}^{k-2}+\ldots +P_{i,j}$,  t=1, 2, …, n.Step 3: The encrypted pixel values *y*^(*t*)^*_i_*_,*j*_ are divided into 8-bits, and each bit is put into the pixels of the shared image *C*^(*t*)^ in order (*C*^(*t*)^*_i_*_,*j*_, …, *C*^(*t*)^*_i_*_,*j*+*k−*1_), and it is called the P part, if k≤7, because k<8 cannot put all the bits at once, so the remaining bits are put in order again and the remaining vacant part is called Part B.Step 4: Repeat Step 2 and Step 3 until all set of pixels (*P_i_*_,*j*_, …, *P_i_*_,*j*+*k−*1_) have been processed.Step 5: Output *n* encrypted shared images *C*^(*t*)^, t=1, 2, …, n.

### 3.2. Data Embedding

In the image embedding part, when the *t_th_* data-hider receives the shared image *C*^(*t*)^, it embeds each pixel of *C*^(*t*)^, and through the B part, it embeds the secret message *S*^(*t*)^ into *C*^(*t*)^ one after another. Figure 6 shows the embedding process of data-hider *t* for the pixel group (*C*^(*t*)^*_i_*_,*j*_, ..., *C*^(*t*)^*_i_*_,*j*+*k−*1_). Since group (*C*^(*t*)^*_i_*_,*j*_, ..., *C*^(*t*)^*_i_*_,*j*+*k−*1_) has a space of 8(*k* − 1) bits, part of the confidential data *S*^(*t*)^ can be directly embedded in this space, resulting in the embedded shared group (*C′*^(*t*)^*_i_*_,*j*_, ..., *C′*^(*t*)^*_i_*_,*j*+*k−*1_). The embedding rate of our method is fixed, and the embedding rate does not decrease as the number of shared images increases. The following is our embedding Algorithm 2.
**Algorithm 2:** Data Embedding
**Input:** Sharing images *C*^(*t*)^, t=1, 2, …, n, Data *S*^(*t*)^**Output:** Marked sharing images *C′*^(t)^, t=1, 2, …, nStep 1: For a pixel *C*^(*t*)^*_i_*_,*j*_ of Sharing images *C*^(*t*)^, the message is taken from the secret message *S*^(*t*)^ and embedded in the B part of pixel *C*^(*t*)^*_i_*_,*j*_. Step 2: Repeat Step 1 until all the B parts of *C*^(*t*)^*_i_*_,*j*_ are embedded in the secret message.Step 3: The output of the embedded shared image *C′*^(t)^.

### 3.3. Data Extracting and Image Decryption

In the part of data extraction and image decryption, the *t_th_* image recipient receives the shared image ${C^{\prime }}^{\left(t\right)}$, and can extract the confidential information *S*^(*t*)^ embedded by the *t_th_* data-hider. In addition, by collecting the shared images and decryption keys above the Threshold, a polynomial can be constructed for each set of pixels, through which the pixels of the original image and the secret information of the image owner can be extracted and the original image *M* can be recovered.

Figure 7 shows the process of extracting the cluster (*C′*^(*t*)^*_i_*_,*j*_, ..., *C′*^(*t*)^*_i_*_,_*_j__+k−_*_1_) by Receiver *t*. The pixel group (*C′*^(*t*)^*_i_*_,*j*_, ..., *C′*^(*t*)^*_i_*_,*j*+*k−*1_) has been embedded in a confidential message of 8 (*k* − 1) bits and can be taken out directly into a partial *S*^(*t*)^. After extraction, there is no need to reduce this group to (*C*^(*t*)^*_i_*_,*j*_, ..., *C*^(*t*)^*_i_*_,*j*+*k−*1_), because the group can obtain *y*^(*t*)^*_i_*_,*j*_ whether it is reduced or not.

Figure 8 shows the response flow diagram of the pixel group (*P_i_*_,*j*_, …, *P_i_*_,*j*+*k−*1_). Assuming that the first *k* shared images are obtained from $C^{\left(1\right)},\;C^{\left(2\right)},\;\ldots ,C^{\left(k\right)}$, the pixels of each shared image are also grouped, and each group contains *k* pixels. For the 7th shared image *C*^(*t*)^ for which the pixel group (*C*^(*t*)^*_i_*_,*j*_, ..., *C*^(*t*)^*_i_*_,*j*+*k−*1_) can be taken out to share pixels *y*^(*t*)^*_i_*_,*j*_. Using *y*^(*t*)^*_i_*_,*j*_, *t* = 1, 2, ..., *k* and $X_{t}$, *t* = 1, 2, ..., *k* and Lagrange polynomials, the originally constructed (*k* − 1)-*th* polynomial can be recovered, and therefore the pixel group (*P_i_*_,*j*_, …, *P_i_*_,*j*+*k−*1_) can be recovered.

The following is our Algorithm 3 for recovering images and extracting secret information. For the sake of illustration, we assume that we obtain the first *k* marked shared images ${C^{\prime }}^{\left(1\right)},\;\ldots ,{C^{\prime }}^{\left(k\right)}$ and use the key $X_{1},\;\ldots ,\;X_{k}$ to decrypt the original images *M* and extract the secret information *S*^(1)^, ..., *S*^(*k*)^.
(4)$$F_{i,j}\left(x\right)=\sum_{t=1}^{k}\left(y^{\left(t\right)}{}_{i,j}\right.\;\prod_{a=1,a\neq t}^{k}\frac{x-X_{a}}{X_{t}-X_{a}})\;$$
where $y^{\left(t\right)}{}_{i,j}$, $X_{t}\epsilon GF\left(256\right),\;t=1,2,\ldots ,k$, and add, subtract, multiply, and divide are all in *GF*(256).
**Algorithm 3:** Data extracting and image decryption
**Input:** Marked sharing images ${C^{\prime }}^{\left(1\right)},\;\ldots ,{C^{\prime }}^{\left(k\right)}$_,_ Decryption keys $X_{1},\;\ldots ,\;X_{k}$**Output:** Data *S*^(1)^, *S*^(2)^,…, *S*^(*k*)^, Original image *M*Step 1. Take part B of the image ${C^{\prime }}^{\left(1\right)},\;\ldots ,{C^{\prime }}^{\left(k\right)}$ to obtain the secret information *S*^(1)^, ..., *S*^(*k*)^.Step 2. For sharing a set of pixels of the image ${C^{\prime }}^{\left(t\right)}$(${C^{\prime }}^{\left(t\right)}$*_i_*_,*j*_, ..., ${C^{\prime }}^{\left(t\right)}$*_i_*_,*j*+*k−*1_), extract the P part of the pixel set and merge it into the encrypted pixel values *y*^(*t*)^*_i_*_,*j*_, t=1, 2, …, k.Step 3. Bring the encrypted pixel values *y*^(1)^*_i_*_,*j,*_*, y*^(*2*)^*_i_*_,*j*_*, ..., y*^(*k*)^*_i_*_,*j*_ with the decryption keys $X_{1},\;\ldots ,\;X_{k}$ into Lagrange polynomial of Equation (4), build the polynomial $P_{i,j+k-1}x^{k-1}+P_{i,j+k-2}x^{k-2}+\ldots +P_{i,j}$.Step 4. The coefficients of the polynomial $P_{i,j+k-1}x^{k-1}+P_{i,j+k-2}x^{k-2}+\ldots +P_{i,j}$ are used to obtain the pixel sets (*P_i_*_,*j*_, …, *P_i_*_,*j*+*k−*1_) of the original images.Step 5. Repeat Step 2 to Step 4 until all pixel groups (*P_i_*_,*j*_, …, *P_i_*_,*j*+*k−*1_) have been processed.Step 6. Combine all pixel groups (*P_i_*_,*j*_, …, *P_i_*_,*j*+*k−*1_) into the original image *M.*Step 7. Output the original image *M* and the secret information *S*^(1),^
*S*^(2),^..., *S*^(*k*)^.

### 3.4. Example

We give an example of Shamir’s (3, 3)-threshold. Assuming the secret information *S*^(1)^= 10000 10000 010000 _(2)_, *S*^(2)^ = 01000 01000 001000 _(2)_, *S*^(3)^= 00100 00100 000100 _(2)_, the pixel values of the pixel group (*P_i_*_,*j*_, …, *P_i_*_,*j*+*k−*1_) of the original image are *P*_(*i,j*)_ = 2 _(10)_*, P*_(*i,j+k*)_ = 4 _(10)_, *P*_(*i,j+*2)_ = 8 _(10)_. The pixel group (*P_i_*_,*j*_, …, *P_i_*_,*j*+*k−*1_) generates 3 pixel groups of shared images after embedding secret information (*C′*^(1)^*_i_*_,*j*_, …, C′^(1)^*_i_*_,*j*+*k−*1_), (*C′*^(2)^*_i_*_,*j*_, …, *C′*^(2)^*_i_*_,*j+k−*1_), (*C′*^(3)^*_i_*_,*j*_, …, *C′*^(3)^*_i_*_,*j+k−*1_).

#### 3.4.1. Encryption and Embedding Steps

The set of pixel values of the original image are *P_(i,j)_* = 2 _(10)_ = 10 _(2)_*, P_(i,j+k)_* = 4 _(10)_ = 100 _(2)_, and *P*_(*i,j+*2)_ = 8 _(10)_ = 1000 _(2)_, respectively. Using the pixel values *P*_(*i,j*)_*, P*_(*i,j+k*)_, and *P*_(*i,j+*2)_ to construct the polynomial $P_{i,j+k-1}x^{k-1}+P_{i,j+k-2}x^{k-2}+\ldots +P_{i,j}$, we obtain $8x^{2}+4x+2$. Then, we start the encryption, at this time we assume the encryption keys *X_t_* are *X*_1_ = 10, *X*_2_ = 20, *X*_3_ = 30, substitute them into $8x^{2}+4x+2$ and get 3 encrypted shared pixels by *GF*(256), respectively, *y*^(1)^ = 60 _(10)_ = **001** 111 00 _(2)_, *y*^(2)^ = 10 _(10)_ = **000** 010 10 _(2)_, *y*^(3)^ = 52 _(10)_ = **001** 101 00 _(2)_. Finally, the encrypted pixel values are divided into 8 bits and stored in the pixel group (*C*^(*t*)^*_i_*_,*j*_, …,*C*^(*t*)^*_i,j+k_*_−1_), t=1, 2, 3, called the P part. We can get

*C*^(1)^*_i_*_,*j*_ = **0**10 00000 _(2)_, *C*^(1)^*_i_*_,*j*+1_ = **0**10 00000 _(2)_, *C*^(1)^*_i_*_,*j*+2_ =**1**1 000000 _(2)_.

*C*^(2)^*_i_*_,*j*_ = **0**01 00000 _(2)_, *C*^(2)^*_i_*_,*j*+1_ = **0**10 00000 _(2)_, *C*^(2)^*_i_*_,*j*+2_ =**0**0 000000 _(2)_.

*C*^(3)^*_i_*_,*j*_ = **0**10 00000 _(2)_, *C*^(3)^*_i_*_,*j*+1_ = **0**00 00000 _(2)_, *C*^(3)^*_i_*_,*j*+2_ =**1**1 000000 _(2)_.

Assuming *S*^(1)^= 10000 **10000** 010000 _(2)_, *S*^(2)^ = 01000 01000 **001000**
_(2)_, *S*^(3)^= **00100** 00100 000100 _(2)_, the data-hider targets the pixels of the three shared images *C*^(1)^, *C*^(2)^, and *C*^(3)^, respectively, the secret information *S*^(1)^, *S*^(2)^, and *S*^(3)^ embedded in part B to obtain the images ${C^{\prime }}^{\left(1\right)}$, ${C^{\prime }}^{\left(2\right)}$, ${C^{\prime }}^{\left(3\right)}$:

${C^{\prime }}^{\left(1\right)}$*_i_*_,*j*_ = 01010000
_(2)_, *C′*^(1)^*_i_*_,*j*+1_ = 010**10000**
_(2)_, *C′*^(1)^*_i_*_,*j*+2_ =11010000 _(2)_.

${C^{\prime }}^{\left(2\right)}$*_i_*_,*j*_ = 00101000
_(2)_, *C′*^(2)^*_i_*_,*j*+1_ = 01001000
_(2)_, *C′*^(2)^*_i_*_,*j*+2_ =00**001000**
_(2)_.

${C^{\prime }}^{\left(3\right)}$*_i_*_,*j*_ = 010**00100**
_(2)_, *C′*^(3)^*_i_*_,*j*+1_ = 00000100 _(2)_, *C′*^(3)^*_i_*_,*j*+2_ =11000100
_(2)_.

#### 3.4.2. Decryption and Extracting Steps

We use 3 shared images to illustrate the steps of decryption and extracting. Take out the pixel sets of 3 shared image images (${C^{\prime }}^{\left(1\right)}$*_i_*_,*j*_, …, ${C^{\prime }}^{\left(1\right)}$*_i_*_,*j*+*k*−1_), (${C^{\prime }}^{\left(2\right)}$*_i_*_,*j,*_ …, ${C^{\prime }}^{\left(2\right)}$*_i_*_,*j*+*k*−1_), (${C^{\prime }}^{\left(3\right)}$*_i_*_,*j*_, …, ${C^{\prime }}^{\left(3\right)}$*_i_*_,*j*+*k*−1_) and the decryption keys *X*_1_, *X*_2_, *X*_3_ and take out each pixel set of B to obtain the secret information *S.* The *P* parts of each pixel group are merged individually to obtain the shared image pixels *y*^(1)^, *y*^(2)^, and *y*^(3)^. Substitute *X*_1_, *X*_2_, *X*_3_, and *y*^(1)^, *y*^(2)^, *y*^(3)^ into Equation (4) to obtain the original polynomial, where *y*^(1)^ = 60, *y*^(2)^ = 10, *y*^(3)^ = 52, *X*_1_ = 10, *X*_2_ = 20, *X*_3_ = 30,
$$60\times \frac{x-20}{10-20}\times \frac{x-30}{10-30}+10\times \frac{x-10}{20-10}\times \frac{x-30}{20-30}+52\times \frac{x-10}{30-10}\times \frac{x-20}{30-20}\;=8x^{2}+4x+2$$

The polynomial $P_{i,j+k-1}x^{k-1}+P_{i,j+k-2}x^{k-2}+\ldots +P_{i,j}$ = 8*x*^2^ + 4*x* + 2 is obtained to obtain the pixel group of the original image (*P_i_*_,*j*_, …, *P_i_*_,*j*+*k*−1_) = (2, 4, 8).

## 4. Experimental Results

In this section, we perform experiments and analysis. All the tested images are gray 425 level sized by 512 × 512. Figure 9 shows the experimental results of Image Boat. We use the three-out-of-four threshold secret sharing method to group every three original images and encrypt them into four shared images. Figure 9a is the original image, Figure 9b–e are different shared images, and Figure 9f is the image after decryption and information retrieval. From the image we can see that our method can fully recover. Similarly, we did the same for image Couple1 and the result is shown in Figure 10.

Figure 11 shows the maximum embedding rate comparison among the proposed method and state-of-the-art methods. We use three-out-of-three, three-out-of-four, and three-out-of-five threshold secret sharing to make the comparison. Here is a three-out-of-four to illustrate that, in our way, a shared image pixel can be embedded with 5 or 6 bits of secret information at an embedding rate of 5.3 bpp (Bit Per Pixel). In Chen et al.’s method, their embedding rate is $\frac{\lfloor \frac{WH-2}{n}\rfloor \;\cdot l}{WH}\approx \frac{l}{n}$ bpp, which is about 1.75 bpp since *n* = 4 and *l* = 7. From Figure 11, we can see that the embedding rate of my method is larger than other methods, and it will be more obvious as the number of shared images increases.

Table 1 shows the feature comparison among the proposed scheme and state-of-the-art schemes, which shows the comparison of the features of our approach with other approaches.

Table 2 shows the comparisons of embedding capacity (bits) and embedding rate (bpp), here the experimental data is used in a three-out-of-four threshold secret sharing approach, in our method, a pixel of a shared image can embed 5 or 6 bits of secret information, so a shared image can embed 1,398,096 bits. The bpp (bit per pixel) for embedding rate is 1,398,096512×512≅ 5.3. We share the image as four shared images, so you can embed 1,398,096×4=5,592,384 bits. Figure 12 shows the images used in Table 2.

Table 3 shows the comparison of embedding rata (bpp) with different *k* based on Shamir (*k*, *n*). In our approach, each original pixel *P_i_*_,*j*_ could be reconstructed by collecting *k* sharing pixels. It means that the *k* sharing pixels holds only *P_i_*_,*j*_ with 8 bits, so *k* sharing pixels has a space of $8\times \left(k-1\right)$ bits, thus the embedding ratio is estimated by 8×k−1k. Assume that k=8 indicates that a pixel *P_i_*_,*j*_ can be split into 8 pixels to produce shared pixels *C*^(1)^*_i_*_,*j*_, …, *C*^(8)^*_i_*_,*j*_ by Shamir’s confidential sharing. Therefore, the embedding ratio is 8×8−18 = 7. In the same cases, assume that k=6 and the embedding ratio is 8×6−16 ≅ 6.6. From Table 3, we can see that our method increases the embedding rate (bpp) as k increases for each k binning of the original images.

If two-out-of-two threshold secret sharing is used for the experiment, the maximum number of entries in the multiplex is only one, which may make the sharing images appear contoured and therefore the encryption effect is not satisfactory. In this case, you just need to add a procedure to encrypt the shared image once more, and then the encrypted image will have the encryption effect. Of course, the decryption process should also add one additional step to the image decryption. We experiment with image Boat. Figure 13a is the original image, Figure 13b,c are the two-out-of-two threshold secret sharing different sharing images, and we can find some contours appear in Figure 13b,c. Figure 13d,e are the results of re-encryption of Figure 13b,c, respectively, which already have the effect of image encryption, while Figure 13d,e are the results of the re-encryption of Figure 13b,c. Figure 13f is the image after decryption and information retrieval.

## 5. Conclusions

This paper proposes an RDHEI technique based on the polynomial construction technique of Shamir’s Secret Sharing technique, which divides *k* pixels into a group and uses *k* pixels to construct a polynomial. Then, our approach uses the polynomial and Galois Field calculation to generate *n* sharing pixels. Because of the finite field ($GF\left(2^{n}\right)$) strategy, our method can avoid the pixel-specific pre-processing. The 8 bits of each shared pixel are split into *k* small parts and placed in each of the *k* pixels of the shared image so that the constructed shared image has B part of the embedding space, and each shared image is encrypted. Information hiders can hide secret information in Part B of the shared image. Compared with other methods, our method has a higher embedding rate, and the embedding rate does not decrease due to more shared images. In the future work, we will consider how to apply the proposed model to other applications or to some specified multimedia such as video or audio.

## Figures and Tables

**Figure 1 entropy-25-00209-f001:**
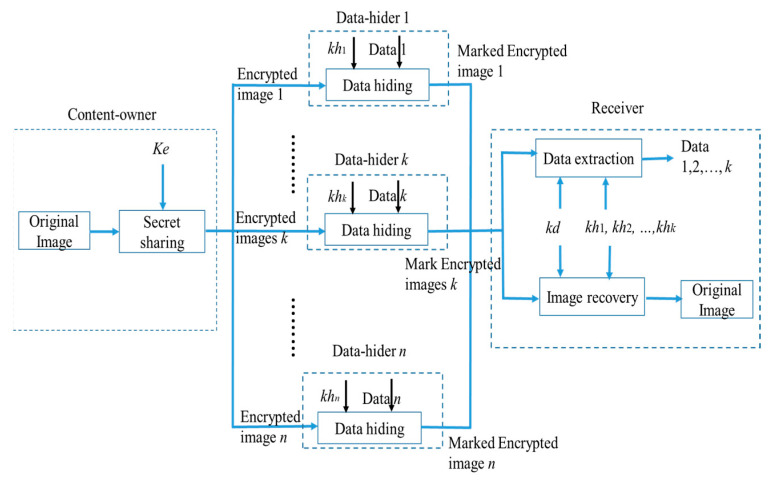
The flowchart of Chen et al.’s method in terms of image encryption, data embedding, data extracting, and image decryption.

**Figure 2 entropy-25-00209-f002:**
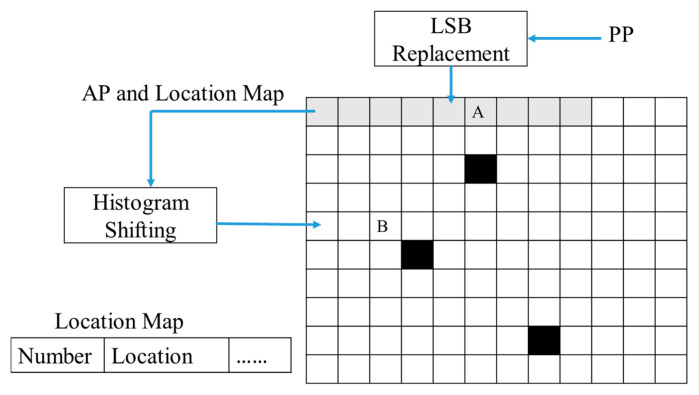
The image is segmented into part A and part B, where black pixels indicate their pixel values ≥ 250.

**Figure 3 entropy-25-00209-f003:**
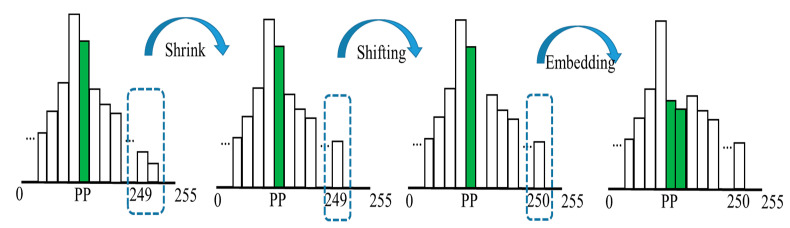
The diagram of pixel value reduction and parameter embedding.

**Figure 4 entropy-25-00209-f004:**
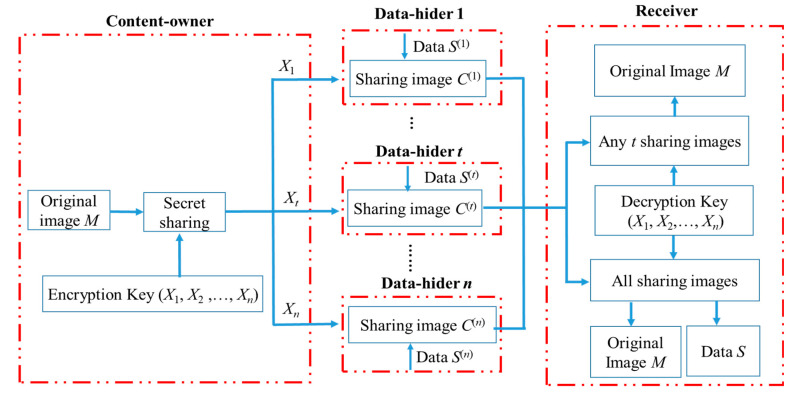
Our proposed diagram in image encryption, data embedding, data extraction and image decryption.

**Figure 5 entropy-25-00209-f005:**
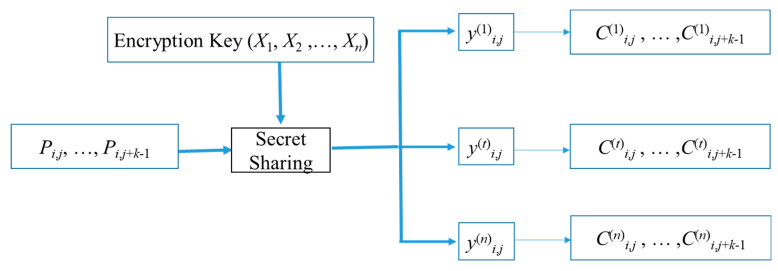
Secret sharing process for pixel group (*P_i_*_,*j*_, …, *P_i_*_,*j*+*k−*1_).

**Figure 6 entropy-25-00209-f006:**
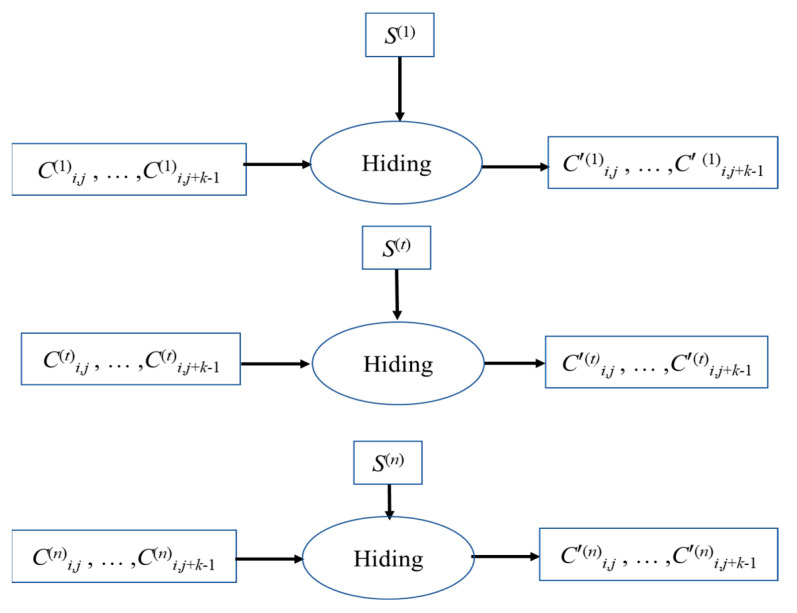
Embedding process for pixel group (*C*^(*t*)^*_i_*_,*j*_, …, *C*^(*t*)^*_i_*_,*j*+*k*−1_), *t* = 1, 2,…, *n*.

**Figure 7 entropy-25-00209-f007:**
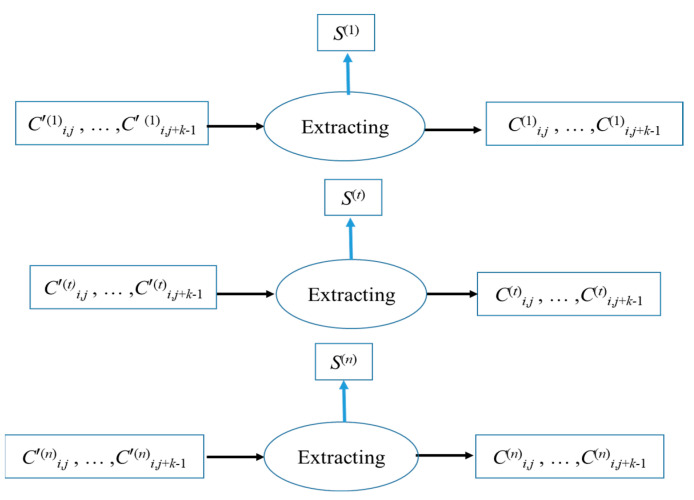
Extraction process for the pixel group (*C′*^(*t*)^*_i_*_,*j*_, ..., *C′*^(*t*)^*_i_*_,*j*+*k*−1_), *t* = 1, 2,…, *n*.

**Figure 8 entropy-25-00209-f008:**
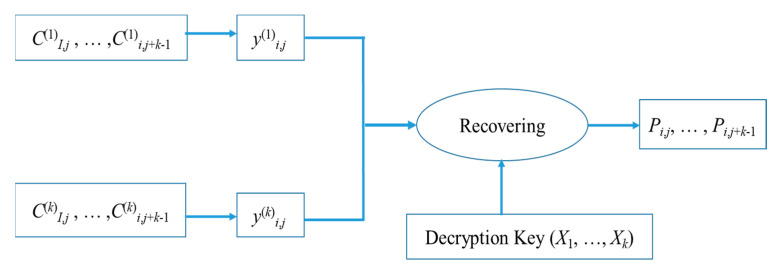
Flowchart of recovery of pixel group (*P_i_*_,*j*_, …, *P_i_*_,*j*+*k*−1_).

**Figure 9 entropy-25-00209-f009:**
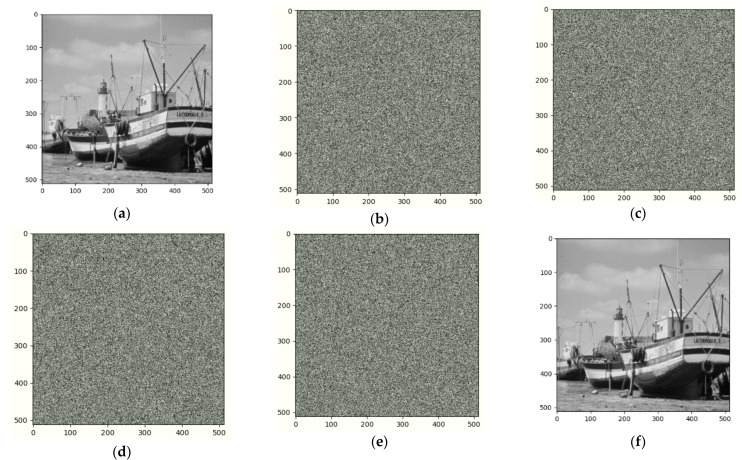
Encrypted results of Boat and its recovered image:(**a**) Original (**b**) Sharing image 1 (**c**) Sharing image 2 (**d**) Sharing image 3 (**e**) Sharing image 4 (**f**) Recovered.

**Figure 10 entropy-25-00209-f010:**
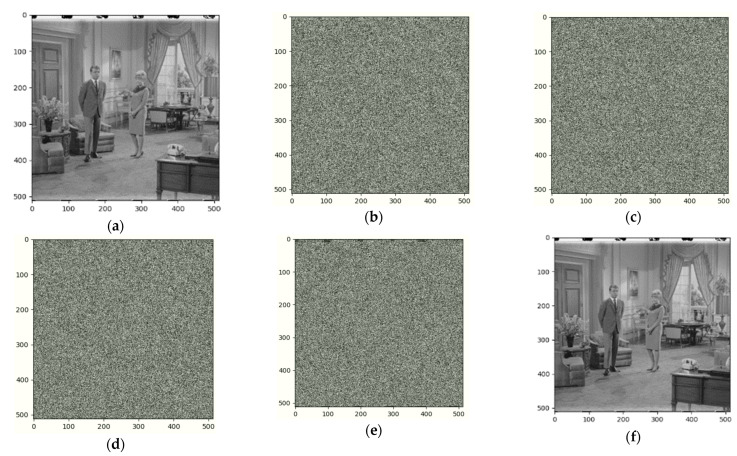
Encrypted results of Couple1 and its recovered image: (**a**) Original (**b**) Sharing image 1 (**c**) Sharing image 2 (**d**) Sharing image 3 (**e**) Sharing image 4 (**f**) Recovered.

**Figure 11 entropy-25-00209-f011:**
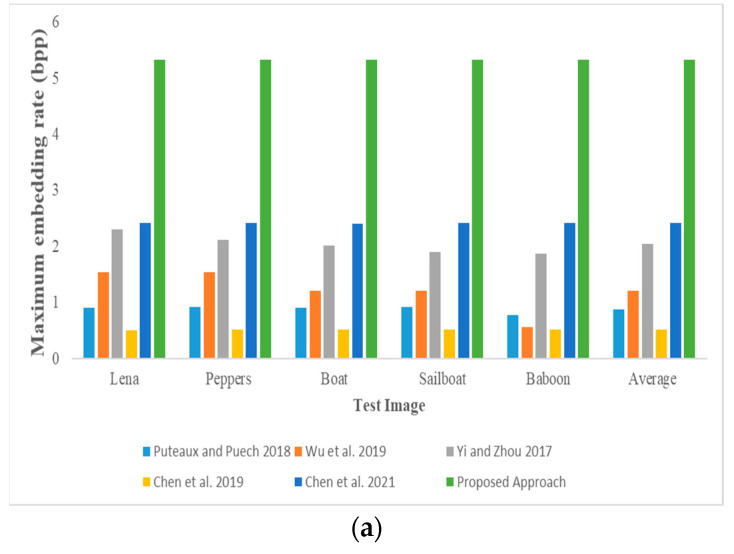
Maximum embedding rate comparison among the proposed method based on Shamir (3, *n*) and state-of-the-art methods [13,14,15,21,22]: (**a**) *n* = 3; (**b**) *n* = 4; (**c**) *n* = 5.

**Figure 12 entropy-25-00209-f012:**
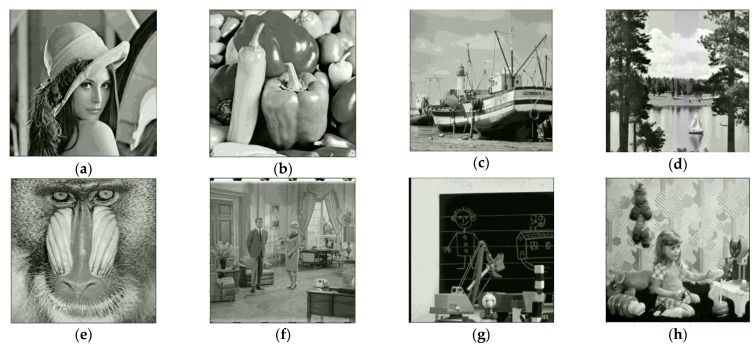
Test images: (**a**) Lena; (**b**) Peppers; (**c**) Boat; (**d**) Sailboat; (**e**) Baboon; (**f**) Couple1; (**g**) Toys; (**h**) Girl.

**Figure 13 entropy-25-00209-f013:**
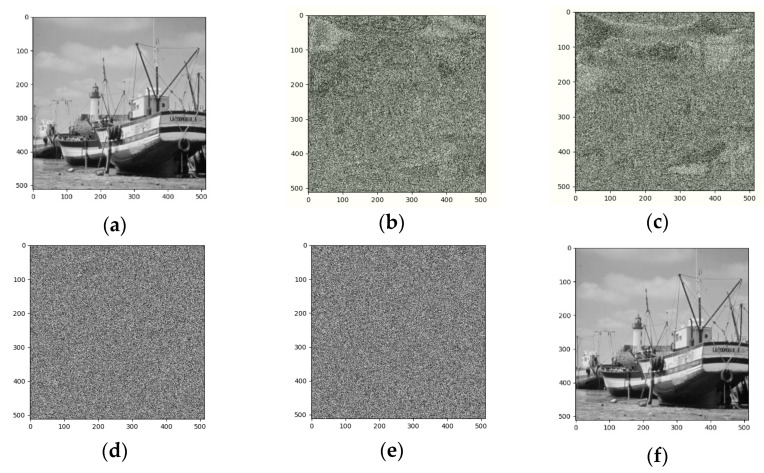
Encrypted results of Boat and its recovered image by 2-out-of-2 threshold secret sharing: (**a**) Original (**b**) Sharing image 1 (**c**) Sharing image 2 (**d**) Encrypted Sharing image 1 (**e**) Encrypted Sharing image 2 (**f**) Recovered.

**Table 1 entropy-25-00209-t001:** Feature comparison among the proposed scheme and state-of-the-art schemes.

Scheme	Separable	Vacating Room before Encryption	Encryption Strategy	Participant (Data-Hider)
Puteaux and Puech [13]	Yes	Yes	Stream cipher	Single
Wu et al. [14]	Yes	Yes	Stream cipher	Single
Yi and Zhou [15]	Yes	Yes	Block permutation and modulation	Single
Chen et al. [21]	No	Yes	Secret sharing	Single
Chen et al. [22]	Yes	No	Secret sharing	Multiple
Proposed	Yes	No	Secret sharing	Multiple

**Table 2 entropy-25-00209-t002:** Comparisons of Embedding capacity (bits) and Embedding rate (bpp).

Test Images	Embedding Capacity (bits)	Embedding Rate (bpp)
Lena	5,592,384	5.3
Peppers	5,592,384	5.3
Boat	5,592,384	5.3
Sailboat	5,592,384	5.3
Baboon	5,592,384	5.3
Couple1	5,592,384	5.3
Toys	5,592,384	5.3
Girl	5,592,384	5.3

**Table 3 entropy-25-00209-t003:** Comparison of Embedding rate (bpp) with different *k* based on Shamir(*k*, *n*).

Test Image	*k*					
	3	4	5	6	7	8
Lena	5.3	6	6.4	6.6	6.8	7
Peppers	5.3	6	6.4	6.6	6.8	7
Boats	5.3	6	6.4	6.6	6.8	7
Sailboat	5.3	6	6.4	6.6	6.8	7
Baboon	5.3	6	6.4	6.6	6.8	7
Average	5.3	6	6.4	6.6	6.8	7

## Data Availability

Not applicable.

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
