# Peer review of "Reversible Data Hiding in Encrypted Image Using Multiple Data-Hiders Sharing Algorithm"

_entropy, 2023, doi:10.3390/e25020209_

Round 1
Reviewer 1 Report
The authors propose the Reversible Data Hiding in Encrypted Image approach based on Shamir's Secret Sharing technique using Galois Field arithmetic. The proposed method can be used in multimedia security technology. The paper contains good background, a theoretical description of the proposed scheme and some practical results of its implementation.
Some comments and suggestions:
1) The authors claim that the proposed method "has a higher embedding rate, and the embedding rate does not decrease due to more shared images". Unfortunately, there is no comparison of the embedding rates of different approaches. Table 1 compares only some features, and one can conclude that the proposed by the authors scheme and Chen's scheme [23] are similar.
Moreover, as far as I understand from the text, the only difference between the proposed approach and Chen's one is using GF(28) operations for the polynomial coefficients. The authors should describe in more detail the benefits of the proposed scheme and give more comparative results of their approach and Chen's one.
2) Table 2 has no practical value because the Embedding capacity and Embedding rate depend on the image size, color depths etc., and not on the image by itself, like "Lena", "Peppers", etc. Almost the same applies to Table 3.
3) The sentence like "Then, we scanned part B, recorded the location of the pixel values ≥ 250 with the location map, modified these pixel values to 249, and created a histogram of part B. From the histogram, we found the" could be a bit confusing due to using pronoun "we". As far as I understand, the authors describe Chen's work, so why are they using "we"?
4) "Fig. 9(a) is the original image, while Fig. 9(b) is the original image." Please correct the sentence.
5) The captures of Fig. 9, 10 and 13 contain references to the parts (b), (c), (d), (e), and (f), while the images themselves – (b), (c), (e), (f), and (g). Please correct.
Reviewer 2 Report
The paper addresses a very important research area and presents an approach based on SSS and multi-project construction. The approach has a clear contribution to the area and is described to a satisfactory level. The approach is evaluated and the results have been well described.
As suggestions for improvements, the following sections would be rewritten to improve readability. The ideas presented are good and correct, but the writing style could be improved. Also, I would recommend updating the conclusion to discuss a bit more weaknesses of the proposed approach and future work in the area.
